# Prevalence of *mecA*- and *mecC*-Associated Methicillin-Resistant *Staphylococcus aureus* in Clinical Specimens, Punjab, Pakistan

**DOI:** 10.3390/biomedicines11030878

**Published:** 2023-03-13

**Authors:** Muhammad Mubashar Idrees, Khadija Saeed, Muhammad Akbar Shahid, Muhammad Akhtar, Khadija Qammar, Javariya Hassan, Tayyaba Khaliq, Ali Saeed

**Affiliations:** 1Institute of Molecular Biology and Biotechnology, Bahauddin Zakariya University, Multan 60800, Punjab, Pakistan; 2Department of Medical Laboratory Technology, The Islamia University of Bahawalpur, Bahawalpur 63100, Punjab, Pakistan; 3Multan Institute of Kidney Diseases (MIKD), Multan 60800, Punjab, Pakistan; 4Department of Pathobiology, Faculty of Veterinary Sciences, Bahauddin Zakariya University, Multan 60800, Punjab, Pakistan; 5Department of Pharmaceutics, Faculty of Pharmacy, The Islamia University of Bahawalpur, Bahawalpur 63100, Punjab, Pakistan; 6Department of Pediatric Oncology & Medical Microbiology, University Medical Center Groningen, University of Groningen, Hanzeplein 1, 9713 GZ Groningen, The Netherlands

**Keywords:** methicillin-resistant *Staphylococcus aureus* (MRSA), antibiotic resistance, *mecC*, *mecA*

## Abstract

Methicillin-resistant *Staphylococcus aureus* (MRSA) is a clinically prevalent bacterium and is resistant to many drugs. Genetic factors such as *mec* genes are considered to be responsible for this resistance. Recently, *Staphylococcal* Cassette Chromosome *mec* (*SCCmec*) element mutations produced *mecC*, a new genetic variant that encodes a transpeptidase enzyme (63% similarity with *mecA*-encoded PBP2a). This cross-sectional study was conducted to establish the prevalence of the *mecA* and *mecC* genes among phenotypically identified MRSA and their effectiveness against different antibiotics in clinical specimens. The prevalence of *Staphylococcus aureus* was 10.2% (*n* = 102) in the total number of clinical specimens collected (*n* = 1000). However, the prevalence of MRSA was 6.3% (*n* = 63) of the total samples collected, while it was 61.8% among total *Staphylococcus aureus* isolates. *mec* genes were confirmed in 96.8% (*n* = 61) isolates of MRSA, while 3.2% (*n* = 2) were found to be negative for *mec* genes. The combination of *mecA* and *mecC* was detected in 57.1% (*n* = 36) of the MRSA isolates. The prevalence of lone *mecA* was 31.8% (*n* = 20) and that of lone *mecC* was 7.9% (*n* = 5) among all the MRSA samples. Penicillin and amoxicillin/clavulanic acid were the most resistant antibiotics followed by norfloxacin (91.2%), levofloxacin (87.1%), ciprofloxacin (83.9%), azithromycin (78.6%), erythromycin (77.4%), moxifloxacin (69.8%), and sulfamethoxazole/trimethoprim (54.9%)**.** On the other hand, vancomycin and teicoplanin (98.4%) were more effective drugs against MRSA followed by linezolid (96.7%), clindamycin (84.6%), chloramphenicol (83.7%), fusidic acid (70.6%), gentamicin (67.7%), and tetracycline (56.8%). In conclusion, a significant prevalence of *mecA* and *mecC* has been found among MRSA isolated from clinical specimens, which is likely responsible for antibiotic resistance in MRSA in our clinical settings. However, vancomycin, teicoplanin, and linezolid were found the top three most effective drugs against MRSA in our clinical settings. Thus, MRSA endemics in local areas require routine molecular and epidemiological investigation.

## 1. Introduction

*Staphylococcus aureus* (*S. aureus*) is a clinically important Gram-positive bacteria found in the normal flora of the skin and nasal cavity. It can cause endocarditis, osteoarthritis, dermal and soft tissue infections, pulmonary and aerobic vaginitis, and even death [1]. *S. aureus* can also cause bloodstream infection, nosocomial pneumonia, and surgical site infection. *S. aureus* is found in 25% of the human population without causing symptoms [2].

Penicillin and penicillin derivatives were extremely effective when used against *Staphylococcal* infections. However, soon after penicillin was approved for clinical use, penicillin-resistant *S. aureus* strains emerged and spread throughout the world. Methicillin was originally introduced for use in clinical settings in 1961; however, the first clinical methicillin-resistant *Staphylococcus aureus* (MRSA) was also reported in the same year in the United Kingdom (UK). MRSA has evolved resistance to a variety of antibiotic classes; it is also known as the “Super Bug” and is a typical multidrug-resistant strain. Before 1990, MRSA was associated with healthcare but, subsequently, distinct strains of MRSA have been reported repeatedly in both humans and livestock [3]. High morbidity and mortality due to infections caused by MRSA are alarming public health concerns. Hospital-acquired MRSA (HA-MRSA) is a major threat to patient safety; therefore, early detection of MRSA is critical for effective infection control [4].

Global surveillance has shown that MRSA has become a threat to clinical settings throughout the world. MRSA is highly prevalent in Europe and the U.S. and is responsible for more than 50% of hospital-acquired infections [5]. The incidence of MRSA is low in the Netherlands and Scandinavian countries due to effective infection control and preventive measures, while MRSA is endemic in most hospitals in Asian countries, especially in developing nations [6].

MRSA strains harbor the *mecA* gene that encodes Penicillin-Binding Proteins 2a (PBP2a), an alternative trans-peptidase with low binding affinity for most β-lactam antibiotics. Therefore, it produces resistance not only to methicillin but also to all members of the extended-spectrum β-lactam antibiotics [7,8]. A novel genetic determinant was recently described, *mecC,* which resulted due to mutations in *mecA*, which encode a trans-peptidase with only 63% identity to *mecA*-encoded PBP2a. *mecC*-containing MRSA isolates belong to cattle and other animals, and they are transferrable from livestock-associated MRSA or other *Staphylococci* to human MRSA [9].

The increasing and continuous resistance of MRSA to many antibiotics may be due to its evolution in genetic factors. It also underlines the importance of animal *Staphylococci* as a reservoir of resistance genes that can potentially contribute to the evolution of antibiotic-resistant human pathogens [10,11]. The administration, prevalence, and importance of the *mecA gene* have been analysed in many studies, but the prevalence of *mecC*-containing MRSA isolates has not been fully understood. In Pakistan, MRSA accounts for a significant proportion of nosocomial infections, and studies have highlighted the increasing clinical importance of MRSA [12]. However, most of these studies have investigated the resistance of MRSA by using phenotypic methods, and limited data are available on genetic and molecular typing of the clinical isolates of MRSA in Pakistan [13]. The main objective of the current study is to establish the antibiotic susceptibility profile of MRSA against different antibiotics and to determine the prevalence of *mecA* and *mecC* genes in MRSA in our clinical settings.

## 2. Materials and Methods

### 2.1. Sample Collection, Transportation, and Preservation

One thousand urine, blood, and routine samples (wound, pus, mouth swab, and abscess of patients) were collected from June 2021 to December 2021, without any regard to age or sex. Samples were taken from patients admitted to Multan Institute of Kidney Diseases Hospital, Multan, Pakistan (a 150-bed single-specialty hospital), from the outpatient department (OPD), emergency (ER), and inpatient department (IPD) departments. Blood samples were obtained in BD culture vials (BD BACTECTM Plus Aerobic/F, Franklin Lakes, NJ, USA), whereas urine and routine samples were collected in sterile containers [14,15].

After collection, the samples were transported to the laboratory while following different protocols, i.e., urine samples were stored on ice packs within 2–4 h of collection, while blood culture vials were held at room temperature for 4–8 h. All samples were processed for analysis as soon as possible after collection and transportation to the Microbiology Department. After processing, urine samples were preserved at 2–8 °C in the refrigerator and the blood culture vials were kept at room temperature for one week. The preserved samples were used for troubleshooting or confirmation of the results if any doubt was found in the findings [15,16].

The Institutional Review Board (IRB) of the Institute of Molecular Biology and Biotechnology (Reference No. 334/A) granted ethical approval for this cross-sectional study, which was designed and conducted at Bahauddin Zakariya University, Multan, from June 2021 to December 2021.

### 2.2. Isolation and Confirmation of Staphylococcus aureus

To isolate *Staphylococcus species*, urine samples and routine samples were cultured on nutrient media and differential media as blood agar (MSA, Oxoid Ltd., Basingstoke Hampshire, UK) and mannitol salt agar (MSA, Oxoid Ltd., Basingstoke Hampshire, UK), respectively, by using a sterile swab. After inoculation, plates were placed at 37 °C in the incubator. The next day bacterial growth was evaluated. If growth was found, then Gram staining and other biochemical tests were performed using a pure and well-isolated colony. If growth was not observed on the first day, the plates were reincubated for overnight incubation. On the second day, these samples were reported as “No Growth” if growth was not seen. Similarly, for blood samples, samples from BD vials were inoculated onto blood and mannitol salt agar plates 3 days after sample collection and incubation of vials at room temperature, following the same steps as in urine and routine samples. According to the morphology of *S. aureus,* the golden yellow colonies from plates were selected for further study [17]. Initial identification of *Staphylococcus aureus* was performed by Gram staining and biochemical testing including catalase and coagulase tests [18]. *S. aureus* was positive for the catalase and coagulase test. Additionally, the molecular confirmation of *S. aureus* was performed by using specific primers of the *nuc* gene for polymerase chain reaction (PCR) (Table 1, Figure 3). After confirmation, pure colonies of *S. aureus* were taken from the mannitol salt agar plates and preserved using the glycerol stock method for further analysis of genetic variants corresponding to MRSA [19].

### 2.3. Phenotypic Identification of MRSA and Antimicrobial Susceptibility Testing

The Clinical Laboratory and Standards Institute (CLSI) 2020 guidelines were followed for both phenotypic identification of MRSA and antimicrobial susceptibility testing (AST) using the Kirby Baur disk diffusion method [23]. In a test tube, one colony from the MSA plate was suspended in 200 μL of 0.9% germ-free normal saline solution. Using a sterile swab, the inoculum was streaked homogeneously on Muller Hinton Agar (MHA, Oxoid, Hampshire, United Kingdom) plates. According to CLSI, MRSA has resistance to cefoxitin (FOX) antibiotic. Thus, phenotypical identification of MRSA was performed by dispensing cefoxitin (FOX) on an MHA plate by using sterile forceps after preparing the lawn of the inoculum. The same procedure was followed for AST against antibiotics, i.e., penicillin (P), amoxicillin/clavulanic acid (AMC), norfloxacin (NOR), levofloxacin (LFX), ciprofloxacin (CIP), azithromycin (AZM), erythromycin (E), moxifloxacin (MXF), sulfamethoxazole/trimethoprim (SXT), tetracycline (TE), gentamycin (G), fusidic acid (FD), chloramphenicol (C), clindamycin (DA), linezolid (LZD), vancomycin (VA), teicoplanin (TEC), and identification of MRSA for each sample. Plates were incubated at 37 °C for 24 h. After 24 h, the zone of inhibition (ZOI) was measured, and the results were interpreted according to the CLSI guidelines. The isolates showing ZOI ≤ 19 mm were characterized as MRSA and with ≥22 as MSSA [24].

### 2.4. DNA Extraction for PCR

The modified CTAB (Cetyltrimethylammonium bromide) method was used for DNA extraction from bacteria [25]. The DNA was extracted from purified (on Tryptic Soya Broth (TSB) for 24 h at 37 °C) and biochemically confirmed *Staphylococcus aureus*. Briefly, 10% Sodium Dodecyl Sulfate (SDS), *Proteinase* K (Thermo Scientific™ *Proteinase* K), 10% CTAB/Sodium chloride (NaCl), and 5M NaCl were used for cell lysis. Extracted DNA was suspended in 100 µL of Tris-Ethylenediaminetetraacetic acid buffer (TE Buffer) and stored at −20 °C. Qualitative measurement of DNA was performed using a 0.5% agarose gel. The Gel Doc system (*BIO-RAD* Gel Doc^TM^ XR + with Image Lab^tm^ Software, BioRad, Hercules, CA, USA) was used for gel visualization.

### 2.5. Molecular Detection of mecA and mecC Genes in MRSA

Antibiotic resistance genes (*mecA* and *mecC*) were amplified by using a basic Thermal Cycler PCR (*BIO-RAD* T100^TM^ Thermal Cycler, Massachusetts, USA) in which specific primers were used for both *mecA* and *mecC* genes (Table 1). A total volume of the PCR reaction (15 µL) contained 2 µL (10 ng/µL) of DNA, 7.5 µL of 2X Taq master mix (Vazyme Biotech Co., Nanjing, China), 1 µL (10 µM) of forward primer, 1 µL (10 µM) of reverse primer, and 3.5 µL of deionized water. PCR for both genes (*mecA* and *mecC* genes) was performed by applying the following conditions: first denaturation at 95 °C for 5 min followed by 35 cycles of 95 °C for 1 min, 55 °C for 30 s, 72 °C for 1 min, and 72 °C for 5 min [21,22]. Molecular detection of *mecA* and *mecC* was performed by loading 6–7 µL of 100 bp DNA ladder into the first well and the remaining wells were loaded by PCR products using 1.5% agarose gel containing 0.5 mg/mL of ethidium bromide in Tris-borate-EDTA (10 mM+1mM EDTA; pH 8.0) buffer used by applying 120 V for an hour. DNA bands were observed under ultraviolet light (UV-light) using the Gel Doc system (*BIO-RAD* Gel Doc^TM^ XR + with Image Lab^tm^ Software, BioRad, Hercules, CA, USA).

### 2.6. Quality Control

*Escherichia coli* (ATCC 25922) and *Staphylococcus aureus* (ATCC 25923) strains from the American Type Culture Collection (ATCC) were used as a control to evaluate the growth-supporting ability of mannitol agar, blood agar, CLED, and MHA agar to maintain the quality of growth throughout the investigation. Gram staining was performed on samples, along with quality control strains of *Staphylococcus aureus* (ATCC 25923) and *Escherichia coli* (ATCC 25922) as a Gram-positive cocci and a Gram-negative rod, respectively. Using ATCC strains of *Staphylococcus aureus* (ATCC 25923) and *Escherichia coli*, the accuracy and reproducibility of biochemical test results were maintained.

### 2.7. Statistical Analysis

In this study, statistical analysis was performed on the data using the *chi*-square test. GraphPad Prism 9 was used for statistical analysis (GraphPad Software, San Diego, CA, USA) and a *p*-value ≤ 0.05 was used to illustrate the statistically significant differences.

## 3. Results

### 3.1. Socio-Demographic Data and Bacterial Distribution among Clinical Patients

A total of one thousand samples were collected and screened. Among these, a total of 309 samples were positive for bacterial growth. After Gram staining, biochemical analysis, and phenotypical identification by disk diffusion method, 10.2% of samples were positive for *Staphylococcus aureus* (*n* = 102) and 6.3% (*n* = 63) were confirmed as MRSA. Among the total MRSA strains (*n* = 63; 61.8%) out of the 102 *Staphylococcus aureus*, MRSA strains isolated from urine, blood, and routine samples (other body fluids, throat swab, pus, abscess, and sputum) accounted for 17.5% (*n* = 11/63), 31.8% (*n* = 20/63,) and 50.8% (*n* = 32/63), respectively. The prevalence of MRSA was slightly higher in male patients, 54% (*n* = 34/63), as compared to female patients, 46% (*n* = 29/63). The MRSA percentages of *S. aureus*, isolated from clinical samples collected from the patients admitted in outpatient department (OPD), inpatient department (IPD), and emergency (ER) wards, were 36.5% (*n* = 23/63), 33.3% (*n* = 21/63), and 30.2% (*n* = 19/63), respectively (Figure 1).

### 3.2. Antibiotic Susceptibility Test

MRSA is characterized mostly by a resistance to β-lactam antibiotics through *mecA*-encoded PBP2a; therefore, MRSA has also shown complete resistance to β-lactam antibiotics including penicillin and amoxicillin/clavulanic acid along with complete resistance against cefoxitin (Figure 2). Most MRSA isolates have reflected resistance to the majority of antibiotics including norfloxacin (91.2%), levofloxacin (87.1%), ciprofloxacin (83.9%), azithromycin (78.6%), erythromycin (77.4%), moxifloxacin (69.8%), and sulfamethoxazole/trimethoprim (54.9%). On the other hand, vancomycin and teicoplanin (98.4%) had more effectiveness against MRSA followed by linezolid (96.7%), clindamycin (84.6%), chloramphenicol (83.7%), fusidic acid (70.6%), gentamicin (67.7%), and tetracycline (56.8%) (Figure 2). Vancomycin, teicoplanin, linezolid, clindamycin, chloramphenicol, fusidic acid, gentamicin, and tetracycline can still be administered in MRSA-associated infections in our clinical settings.

### 3.3. Prevalence of mecA and mecC Genes in MRSA

*Staphylococcus aureus* was isolated and confirmed as MRSA in 63 (61.8%) out of the total 102 *S. aureus* isolates by phenotypic and biochemical methods. Further, PCR was performed to check the prevalence of *mecA* and *mecC* genes in MRSA. The amplicon sizes of *mecA* and *mecC* genes were 533 bp and 356 bp, respectively (Figure 3).

A total of fifty-six (*n* = 56) isolates were confirmed for *mecA* with PCR from total isolates of MRSA (*n* = 63). The prevalence of the *mecA* gene was 88.8% (*n* = 56/63), but the *mecA* gene was not detected in seven isolates (*n* = 7/63; 11.1%). Remarkably, these isolates showed complete resistance to cefoxitin and penicillin in the antibiotic sensitivity test. The *mecC* gene was identified in forty-one isolates (*n* = 41/63; 65.0%) and the remaining twenty-two showed no amplification (*n* = 22/63; 34.9%).

Sixty-one isolates (*n* = 61/63; 96.8%) were positive for *mec* genes, while two isolates (*n* = 2/63; 3.2%) were found to be negative for any *mec* gene. Both *mecA* and *mecC* genes were found in thirty-six isolates (*n* = 36/63; 57.1%); this is a higher prevalence than the *mecA* gene alone (*n* = 20/63; 31.8%) and the *mecC* gene alone (*n* = 5/63; 7.9%) (Figure 4).

## 4. Discussion

Infections with MRSA are prevalent in both healthcare facilities and the general population. *mecC* is an emerging gene responsible for Staphylococcal methicillin resistance. The antimicrobial susceptibility pattern of MRSA isolates having *mecA* and *mecC* genes differs from non-MRSA isolates, which frequently exhibit resistance to both penicillin and cefoxitin antibiotics. In contrast, the majority of genetic factors (*mecA* and *mecC* gene) containing MRSA are resistant to penicillin and cefoxitin and are subsequently reported as MRSA, and may be susceptible to remaining antibiotics [26].

In this study, we reported the prevalence of *mecA* and *mecC* genes in MRSA isolated from our local population of southern Punjab, Pakistan. In addition, we also conducted a study on the antibiotic resistance profile of MRSA against antibiotics other than penicillin and cefoxitin. Several studies reported antibiotic resistance of MRSA in this region; however, the genetic factor responsible for resistance was not broadly studied in our clinical settings. In this study, *S. aureus* was isolated from routine clinical samples including blood, urine, sputum, pus, and body fluids collected from different hospital wards. A high incidence (10.2%) of *S. aureus* in routine clinical samples indicates that *S. aureus* infections are a major contributor to infections in our clinical settings as compared to other Gram-positive and Gram-negative bacteria [27]. Furthermore, the prevalence of MRSA was 61.8% in total *S. aureus* isolates, while methicillin-susceptible *S. aureus* (MSSA) was confirmed in 38.2%. Similar to the previous study, all MRSA isolates were also found to be resistant to the β-lactam antibiotics including cefoxitin, amoxicillin, and penicillin [28]. In contrast to previous studies in Pakistan [29,30], MRSA isolates were found to be more susceptible to non-β-lactam antibiotics, i.e., fusidic acid, gentamicin, tetracycline, clindamycin, and chloramphenicol in this study. Previously, MRSA isolates were shown to be more resistant to non-β-lactam antibiotics, i.e., fusidic acid, tetracycline, clindamycin, and chloramphenicol. This is likely due to less use of these antibiotics in past in our clinical patients, which may allow MRSA to restore sensitivity against these drugs.

In this study, MRSA isolates were highly resistant to erythromycin, norfloxacin, ciprofloxacin, levofloxacin, and azithromycin, which is in line with previously conducted studies in this region [31,32]. This indicates overuse of antibiotics in this area, or an evolutionary adaptation of bacteria [33]. Vancomycin and teicoplanin have already exhibited some (1.6%) resistance in our clinical settings. This is an alarming situation, as intermediate or resistant strains are already emerging in this area; vancomycin is considered to be the only effective drug against MRSA infections. A recently launched antibiotic called linezolid is being used to treat MRSA infections as well. It can bind to the 23S ribosomal RNA of the larger subunit of the bacterial ribosome and inhibits protein synthesis [34]. It is also a remarkable finding in this study that 3% of MRSA isolates are already showing resistance against linezolid. This obviates the importance of continuous monitoring of drug resistance in all clinical settings where the use of antibiotics is a common clinical practice.

Generally, the *mecA* gene has been considered the most prevalent in MRSA compared to *mecC*. For example, the prevalence of *mecA* has been reported at around 31.9% in MRSA [35,36]. Similarly, in this study, the prevalence of *mecA* alone was 31.8%, and the prevalence of *mecC* alone was 7.9% in MRSA. Remarkably, the prevalence of the *mecA* and *mecC* combination was the highest at 57.1% in MRSA. This indicates an accumulative effect of both genetic factors in the incidence of MRSA, which is an alarming situation for clinical settings. A similar incidence of *mecA* (100%; *n* = 50) and a combination of both *mecA* and *mecC* (6%; *n* = 3/50) has recently been reported for MRSA in Egypt in 2020 [37]. A decade before this, a low incidence (0.45%) of the *mecC* gene in MRSA was reported in other countries [38]. The *mecC* gene in MRSA was first reported in our hospital settings in 2020 [39]. Remarkably, we are reporting a distressing increase in the prevalence of *mecC* in MRSA in our clinical settings within recent years [39]. A previous study from Pakistan reflects a similar situation, with MRSA containing both *mecA* and *mecC* genes [39]; however, the prevalence of this combination reported in this study was lower. Further, the *mecA* and *mecC* genes were not detected in two clinical isolates in this study, although these were phenotypically and biochemically characterized as MRSA. This may indicate the need for other molecular confirmations along with conventional methods to characterize MRSA, as previously described [39,40]. Furthermore, it is also possible that another mutant may be circulating in our hospital-acquired MRSA. Further studies are still required to characterize these isolates.

The origin of MRSA-carrying *mecC* is not yet clear, but this novel gene has also been reported in other *Staphylococcus* species. [41]. This suggests that coagulase-negative *Staphylococcus* spp. may be the source of *mecC* in MRSA, as previously reported for *mecA* as well [42]. Thus, the genetic transformation from *Staphylococci* to *S. aureus* may play a pivotal role in increasing the incidence and prevalence of *mecC* in *S. aureus* in the last couple of years in our clinical settings. Therefore, clinical microbiologists should be aware of the risk of *mecC* transformation from other methicillin-resistant Staphylococcal pathogenic species [43].

However, *mecC* remained more prevalent in Denmark than in other countries of Europe and the subcontinent. Many cases of *mecC*-mediated MRSA were reported in Spain, especially in skin infections that supposedly emerge from livestock [28]. Now, countries in Asia have also reported cases of MRSA with a prevalence of the *mecC* gene [44]. The MRSA cases in this study are clinical isolates and also appeared to be multi-drug-resistant. These results indicate that these MRSA cases may carry an SCC *mec* element of type III [45]. SCC *mec* element IV is present in community-associated MRSA and they do not resist multiple antibiotics. The incidence of *mecC* in MRSA may have an impact on the multidrug resistance quality and make them difficult to treat with new antimicrobial agents. *mecC*-carrying MRSA harbors other antibiotic-resistant genes and their regulator genes for expression. We showed the resistance profile of clinical MRSA isolates for many antibiotics. The antibiotic resistance rate for non-β lactams antibiotics was low. While it may be possible that there was an absence or no expression of the genetic factors responsible for those antibiotics, non-β-lactam antibiotics may be used for MRSA infections in this area.

## 5. Conclusions

We concluded that the *mecA* gene was more prevalent compared to the *mecC* gene in MRSA isolated from our clinical settings. However, the prevalence of the *mecC* gene is increasing gradually. Vancomycin, linezolid, teicoplanin, chloramphenicol, and clindamycin can be used against infections caused by MRSA due to their lower resistance rate. The molecular investigation of the *mecA* and *mecC* genes in MRSA should be a routine practice in our clinical settings. The routine molecular epidemiology of *mecC* should be conducted using other clinical isolates so we can find ways to avoid the genetic transformation of *mecC* into *mecC*-negative bacterial species. The changing molecular basis of MRSA drug resistance affects not only new treatment strategies for MRSA but also impedes the control of MRSA infections. There is a need for more molecular and epidemiological investigations to prevent the spread of *S. aureus* in local areas and to understand the rise of multi-drug resistance in MRSA.

## Figures and Tables

**Figure 1 biomedicines-11-00878-f001:**
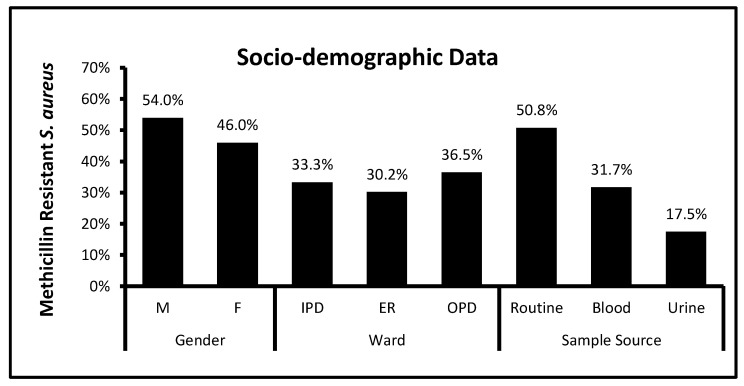
Socio-demographic data of clinically isolated MRSA. Socio-demographic data of clinically isolated MRSA contained a type of samples, wards, and sex. M = male, F = female, IPD = in patient, ER = emergency, OPD = outpatient.

**Figure 2 biomedicines-11-00878-f002:**
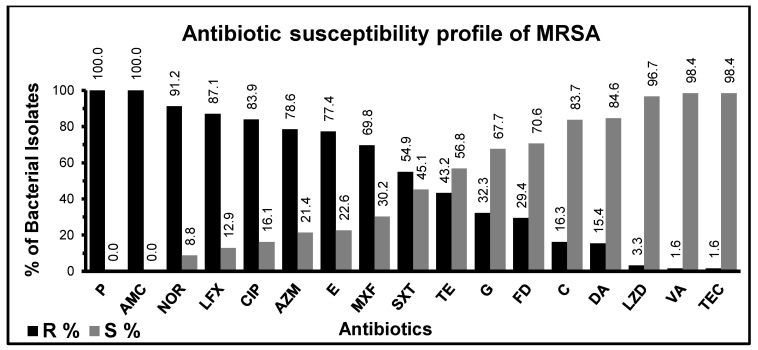
Antibiotic resistance and susceptibility pattern in MRSA. Resistance pattern of MRSA (characterized by Cefoxitin) for penicillin (P) amoxicillin/clavulanic acid (AMC), norfloxacin (NOR), levofloxacin (LFX), ciprofloxacin (CIP), azithromycin (AZM), erythromycin (E), moxifloxacin (MXF), sulfamethoxazole/trimethoprim (SXT), tetracycline (TE), gentamycin (G), fusidic acid (FD), chloramphenicol (C), clindamycin (DA), linezolid (LZD), vancomycin (VA), and teicoplanin (TEC). The percentage of resistance is shown by the dark black column and the percentage of susceptibility is shown by the light grey column. The prevalence of AST is significantly associated with MRSA having a *p*-value < 0.05.

**Figure 3 biomedicines-11-00878-f003:**

Molecular detection and prevalence of resistance-associated genetic factors in MRSA. (**a**) Lane-PC stands for positive control, and Lane-1, -2, -3, -4, -5, -6, -7, and -8 show bands of 356 bp, which is related to the *mecC* gene. (**b**) Lane PC stands for positive control and Lane-1, -2, -3, -4, -5, -6, -7, and -8 show bands of 533 bp, which is related to the *mecA* gene. (**c**) Lane PC and NC stand for positive control and negative control, respectively; Lane-1, -2, -3, -4, -5, -6, -7, -8, and -9 show bands of 279 bp, which is related to the *nuc* gene.

**Figure 4 biomedicines-11-00878-f004:**
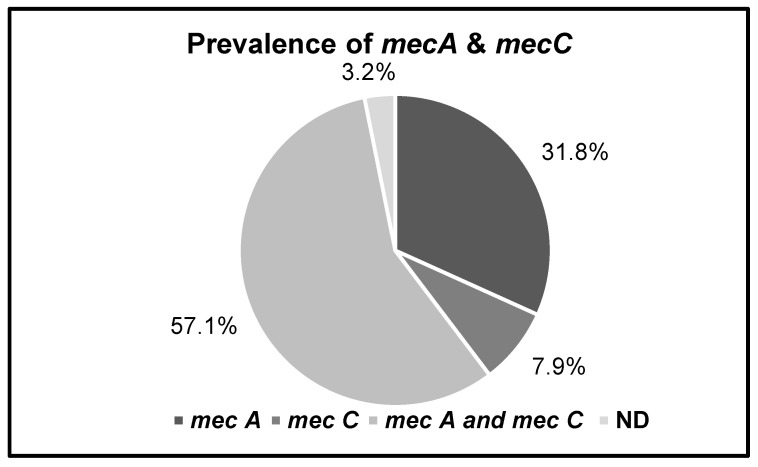
Prevalence of genetic variants (*mecA* and *mecC*) in MRSA. The *mecA* and *mecC* were detected by PCR analysis in MRSA isolated from clinical specimens (routine samples, i.e., throat swab, pus, abscess, sputum, body fluids, blood, and urine) that were significantly associated with MRSA (*p*-value = 0.000). ND = not detected.

**Table 1 biomedicines-11-00878-t001:** Primer used for identification of *S. aureus* and genetic variants of MRSA in this study.

Target Gene	Primer Sequence (5′ to 3′)	Fragment Size (bp)	Reference
** *nuc ** **	F -GCGATTGATGGTGATACGGTI- R -AGCCAAGCCTTGACGAACTAAAGC-	279 bp	[20]
** *mecA* **	F -AAAATCGATGGTAAAGGTTGGC- R -AGTTCTGGAGTACCGGATTTGC-	533 bp	[21]
** *mecC* **	F -TCACCAGGTTCAAC[Y]CAAAA- R -CCTGAATC[W]GCTAATAATATTTC-	356 bp	[22]

* The list of primers used in this study is mentioned in this table. *nuc* gene primers were used for molecular confirmation of *S. aureus,* while the other two pairs of primers (*mecA* and *mecC*) were used for the detection of genetic variants and confirmation of molecular mechanisms of drug resistance.

## Data Availability

Not applicable.

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
