# Peer review of "Prevalence of mecA- and mecC-Associated Methicillin-Resistant Staphylococcus aureus in Clinical Specimens, Punjab, Pakistan"

_biomedicines, 2023, doi:10.3390/biomedicines11030878_

Round 1

Reviewer 1 Report

There are certain concerns that need to be explored in relation to the article I reviewed.

First of all, I think it's urgent to fix Figure 2.

In particular, I saw that the antibiotic "amikacin" is reported in the caption of the figure. However, it is quoted just once in the article to identify MRSA strains, and its data are not part of the graph.

On the contrary, the graph shows data referred to as “TEC”, which I assume stands for "teicoplanin”, whose effectiveness is discussed in the text even though the antibiotic is not explicitly quoted in the caption below the figure.

I suggest the authors update the graph and the caption where necessary to avoid inconsistencies among the antibiotics used and the related data reported.

Another issue can be found on page 5, where there seems to be a missing word, most probably a name judging by the context, between the words “including” and “followed”. 

An apparent mistake was found approximately at the center of the seventh page, where it is stated that the prevalence of the combination of the genes mecA and mecC is 31.7%, such percentage being claimed as the highest: actually, previous Figure 4 had already shown that the proper value should be 57.1%.

Talking about percentages, the authors’ choice of using 61.7% to express, from page 4 onward, the prevalence of MRSA in the sample appears a bit odd. considering that they had stated that the MRSA strains were 63 out of the 102 samples positive for Staphylococcus aureus, which leads to a percentage of 61.764, which should have been rounded up to 61.8%. Also Figure 4 shows improper use of rounding since the total amount of the data sums up to 99.9% only.

The word "carefully" is used to describe the sample transportation method at the beginning of the essay, which is notable. While there is no reason to question the flawless execution of the transportation, it would not be harmful to provide further details regarding the protocols that were followed.

Also, further down in the article, there is an observation made by the authors that can be found confusing, as I believe there is not enough evidence to state it. I’m referring to the allusion to the fact that the bacteria in this study contains SCCmec type III genes: while this is certainly possible, and it is in fact stated in other articles that SCCmecIII bacteria are typically HA-MRSA and multidrug-resistant, the article they cite only alludes to their own population and the prevalence of this particular type in their own samples.

Since the sample in this study came from a hospital context, and since the bacteria are multidrug-resistant, it is plausible to think that the authors are correct in defining them as -type III, however, I couldn’t find any sequencing in the article, and so I believe is for the best not to give it as a certainty that they are.

Author Response

Response to Reviewer 1:

Sr.No

Question/Query

Answer/comment

1.       

There are certain concerns that need to be explored in relation to the article I reviewed. First of all, I think it's urgent to fix Figure 2.

Thank you! Figure 2 is revised in the new version of the manuscript.

2.       

In particular, I saw that the antibiotic "amikacin" is reported in the caption of the figure. However, it is quoted just once in the article to identify MRSA strains, and its data are not part of the graph.

This is removed from the caption of Figure 2 and section 2.3 in the revised manuscript.

3.       

On the contrary, the graph shows data referred to as “TEC”, which I assume stands for "teicoplanin”, whose effectiveness is discussed in the text even though the antibiotic is not explicitly quoted in the caption below the figure.

Teicoplanin (TEC) is mentioned now in the caption of figure 2 as “Figure 2. Antibiotic resistance and susceptibility pattern in MRSA. Resistance pattern of MRSA (characterized by Cefoxitin) for penicillin (P) amoxicillin/clavulanic acid (AMC), norfloxacin (NOR), levofloxacin (LFX), ciprofloxacin (CIP), azithromycin (AZM), erythromycin (E), moxifloxacin (MXF), sulfamethoxazole/trimethoprim (SXT), tetracycline (TE), gentamycin (G), fusidic acid (FD), chloramphenicol (C), clindamycin (DA), linezolid (LZD), vancomycin (VA), and teicoplanin (TEC) a percentage of resistance is shown by dark black column and susceptibility by light grey column”.

4.       

I suggest the authors update the graph and the caption where necessary to avoid inconsistencies among the antibiotics used and the related data reported.

We revised according to the reviewer’s suggestion.

5.       

Another issue can be found on page 5, where there seems to be a missing word, most probably a name judging by the context, between the words “including” and “followed”. 

We updated this in the revised manuscript as

“Most of the MRSA samples have reflected resistance to the majority of antibiotics including norfloxacin (91.2%), levofloxacin (87.1%), ciprofloxacin (83.9%), azithromycin (78.6%), erythromycin (77.4%), moxifloxacin (69.8%) and sulfamethoxazole/trimethoprim (54.9%)”.

6.       

An apparent mistake was found approximately at the center of the seventh page, where it is stated that the prevalence of the combination of the genes mecA and mecC is 31.7%, the such percentage being claimed as the highest: actually, previous Figure 4 had already shown that the proper value should be 57.1%.

This is also updated:

“Remarkably, the prevalence of mecA and mecC combination was the highest 57.1% in MRSA”.

7.       

Talking about percentages, the authors’ choice of using 61.7% to express, from page 4 onward, the prevalence of MRSA in the sample appears a bit odd. considering that they had stated that the MRSA strains were 63 out of the 102 samples positive for Staphylococcus aureus, which leads to a percentage of 61.764, which should have been rounded up to 61.8%. Also Figure 4 shows improper use of rounding since the total amount of the data sums up to 99.9% only.

Replaced 61.7% with 61.8% in the whole manuscript and also updated the data sums up to 100% in Figure 4.

8.       

The word "carefully" is used to describe the sample transportation method at the beginning of the essay, which is notable. While there is no reason to question the flawless execution of the transportation, it would not be harmful to provide further details regarding the protocols that were followed.

Section 2.1 has been modified as:

“One thousands of urine, blood, and routine samples (wound, pus, mouth swab, and abscess of patients) were collected from June 2021 to December 2021, without any regard to age or gender. Samples were taken from patients admitted to Multan Institute of Kidney Diseases Hospital, Multan, Pakistan (a 150-bed single-specialty hospital), from the out-patients door (OPD), emergency (ER), and indoor patient (IDU). Blood samples were obtained in BD culture vials (BD BACTECTM Plus Aerobic/F, Franklin Lakes, NJ, USA), whereas urine, and routine samples were collected in a sterile container [12,13].

After collection, the samples were transported to the laboratory following different instructions i.e., urine samples within 2 to 4 hours placed on ice packs and blood culture vials held at room temperature for 4 to 8 hours. Urine samples were kept at 2 °C to 8 °C in a refrigerator after processing, and blood culture vials were kept at room temperature for one week [13,14].

The Institutional Review Board (IRB) of Institute of Molecular Biology and Biotechnology (Reference No. 334/A) granted ethical approval for this cross-sectional study, which was designed and conducted at Bahauddin Zakariya University, Multan, from June 2021 to December 2021”.

9.       

Also, further down in the article, there is an observation made by the authors that can be found confusing, as I believe there is not enough evidence to state it. I’m referring to the allusion to the fact that the bacteria in this study contains SCCmec type III genes: while this is certainly possible, and it is in fact stated in other articles that SCCmecIII bacteria are typically HA-MRSA and multidrug-resistant, the article they cite only alludes to their own population and the prevalence of this particular type in their own samples.

Since the sample in this study came from a hospital context, and since the bacteria are multidrug-resistant, it is plausible to think that the authors are correct in defining them as -type III, however, I couldn’t find any sequencing in the article, and so I believe is for the best not to give it as a certainty that they are

Thanks for this comment, we agree with the reviewer. It was mentioned on assumption by comparing our clinical data with animal data on MRSA. Sequencing was not aimed in this study. This statement was added for discussion, which is changed now.

 “MRSA in this study are clinical isolates and also appeared as multi-drug resistant. So these results indicated that these MRSAs may carry SCC mec element of type III [45].

Reviewer 2 Report

I found this article interesting for the readers of journal Biomedicines and followed the journal’s scope. I don’t have any major comments as presentation of the research work with data and proper discussion have made this article more interesting to the reader of Biomedicines.

I would recommend the article be published in Biomedicines after minor corrections. 

The author needs to address the following comments/corrections.

 1.    The author should correct the format of references (7 and 27) wherever needed (e.g Year Bold, Volume Italic, pp etc.).

2.    The author should include the full form of abbreviation when first used in the text (e.g SCC).

3.    For 2.1. Sample collection: The author should mention the storing condition of the samples.

4.    For “In total one thousand samples were collected of urine, blood, wound, pus, mouth swab, and abscess of patients…”: The author could include the numbers of different samples in the figure.

5.    Includes footnotes for Table 1.

6.    For all “μl”: Change to “μL”.

7.    For all “37°C”: Change to “37 °C (space).

8.    Figure 1: Include emergency (ER) in the footnotes.

9.    All figures and tables should have a title.

10.  Figure 4: The author could present the dt in figure 4 as a bar graph like other figures.

11. Font of mecC in the sentence needs to change: “However, clinical microbiology laboratories should be aware of mecC not only in MRSA, but they should aware of its occurrence of mecC in other pathogenic species of methicillin-resistant Staphylococci [32].”

12. The author could include the following relevant references:

(a)   Shebl HR, Zaki WK, Saleh AN, Salam SAA. Prevalence of MecC Gene Among Methicillin Resistant Staphylococcus aureus isolated from Patients in Ain-Shams University Hospital. J Pure Appl Microbiol. 2020;14(4):2807-2813. doi: 10.22207/JPAM.14.4.56 

 (b)  Paterson GK, Morgan FJ, Harrison EM, Cartwright EJ, Török ME, Zadoks RN, Parkhill J, Peacock SJ, Holmes MA. Prevalence and characterization of human mecC methicillin-resistant Staphylococcus aureus isolates in England. J Antimicrob Chemother. 2014 Apr;69(4):907-10. doi: 10.1093/jac/dkt462. Epub 2013 Nov 27. PMID: 24284779; PMCID: PMC3956372.

Author Response

Response to Reviewer 2:

S.No.

Question/Query

Answer/comment

1.       

I found this article interesting for the readers of journal Biomedicines and followed the journal’s scope. I don’t have any major comments as presentation of the research work with data and proper discussion have made this article more interesting to the reader of Biomedicines.

I would recommend the article be published in Biomedicines after minor corrections. 

The author needs to address the following comments/corrections.

 1.    The author should correct the format of references (7 and 27) wherever needed (e.g Year Bold, Volume Italic, pp etc.).

Modified the format in this revised manuscript.

2.       

The author should include the full form of abbreviation when first used in the text (e.g SCC).

According to the suggestion of reviewer, we added full name now;

“Recently, Staphylococcal Cassette Chromosome mec (SCCmec) element mutations produced mecC, a new genetic variant that encodes a transpeptidase enzyme (63% similarity with mecA-encoded PBP2a)”.

3.       

For 2.1. Sample collection: The author should mention the storing condition of the samples.

Section 2.1 has been modified as:

“One thousands of urine, blood, and routine samples (wound, pus, mouth swab, and abscess of patients) were collected from June 2021 to December 2021, without any regard to age or gender. Samples were taken from patients admitted to Multan Institute of Kidney Diseases Hospital, Multan, Pakistan (a 150-bed single-specialty hospital), from the out-patients door (OPD), emergency (ER), and indoor patient (IDU). Blood samples were obtained in BD culture vials (BD BACTECTM Plus Aerobic/F, Franklin Lakes, NJ, USA), whereas urine, and routine samples were collected in a sterile container [12,13].

After collection, the samples were transported to the laboratory following different instructions i.e., urine samples within 2 to 4 hours placed on ice packs and blood culture vials held at room temperature for 4 to 8 hours. Urine samples were kept at 2 °C to 8 °C in a refrigerator after processing, and blood culture vials were kept at room temperature for one week [13,14].

The Institutional Review Board (IRB) of Institute of Molecular Biology and Biotechnology (Reference No. 334/A) granted ethical approval for this cross-sectional study, which was designed and conducted at Bahauddin Zakariya University, Multan, from June 2021 to December 2021”.

4.       

For “In total one thousand samples were collected of urine, blood, wound, pus, mouth swab, and abscess of patients…”: The author could include the numbers of different samples in the figure.

These numbers are mentioned in the text as;

“The MRSA strains were 63 (61.8%) out of the 102 samples positive for Staphylococcus aureus, isolated from routine clinical samples from urine (n=11/63, 17.5%), blood (n=20/63, 31.8%) and other body fluids (n=32/63, 50.8%) collected from including throat swab, pus, abscess, sputum”. However, we add only percentages in the figures to avoid the crowdedness. If the reviewer will further point out a figure, where I should also add numbers. We can include it. 

5.       

Includes footnotes for Table 1.

The footnote of Table 1 is mentioned now as:

“We used above mentioned primers in MRSA, nuc primer was used for confirmation of S. aureus while the other two primers of mecA and mecC were for detection of variations”.

6.       

For all “μl”: Change to “μL”.

Thanks for this suggestion also, we changed it now as;

“A total volume of PCR reaction (15 µL) contained 2 µL (10 ng/µL) of DNA, 7.5 µL of 2XTaq master mix (Vazyme Biotech Co., Nanjing, China), 1 µL (10 µM) of forward primer, 1 µL (10 µM) of reverse primer, and 3.5 µL of deionized water. PCR was performed by applying the following conditions: first denaturation at 95 °C for 5 minutes followed by 35 cycles of 95 °C for 1 minute, 55 °C for 30 seconds, 72 °C for 1 minute, and 72 °C for 5 minutes. Molecular detection of mecA and mecC was done by loading 6-7 µL of 100 bp DNA ladder into the first well and the remaining were loaded by PCR product in which 1.5% agarose gel containing 0.5 mg/ml of ethidium bromide in Tris-borate-EDTA (10mM+1mMEDTA; pH=8.0) buffer used by applying 120 V for an hour”.

7.       

For all “37°C”: Change to “37 °C (space).

Thanks for this comment also, we corrected it in the new version.

8.       

Figure 1: Include emergency (ER) in the footnotes.

Thanks, this is added in footnote now as;

“Figure 1. Socio-demographic data of clinically isolated MRSA. Socio-demographic data of clinically isolated MRSA contained type of samples, wards, and gender. *M=male, F=female, IPD=in-door patient, ER=emergency, OPD=out-door patient”.

9.       

All figures and tables should have a title.

Thanks, we added all titles to all graphs and tables.

10.    

Figure 4: The author could present the dt in figure 4 as a bar graph like other figures.

According to suggestion to the reviewer, we make a bar graph, you can have look on this graph, but we still feel pie graph looks better as compared to this. But if review still think this should be change with bar graph, we will add this graph instead of pie graph.

11.    

Font of mecC in the sentence needs to change: “However, clinical microbiology laboratories should be aware of mecC not only in MRSA, but they should aware of its occurrence of mecC in other pathogenic species of methicillin-resistant Staphylococci [32].”

Thanks for this indications; this paragraph is changed as;

“The origin of MRSA-carrying mecC is not yet clear, but this novel gene has also been reported in other Staphylococcus species. [41]. This suggests that coagulase-negative Staphylococcus spp. may be the source of mecC in MRSA, as previously reported for mecA as well [42]. Thus, the genetic transformation from Staphylococci to S. aureus may play a pivotal role to increase the incidence and prevalence of mecC in S. aureus in the last couple of years in our clinical settings. Therefore, clinical microbiologists should be aware of the risk of mecC transformation from other methicillin-resistant Staphylococcal pathogenic species [43].”

12.    

The author could include the following relevant references:

(a)   Shebl HR, Zaki WK, Saleh AN, Salam SAA. Prevalence of MecC Gene Among Methicillin Resistant Staphylococcus aureus isolated from Patients in Ain-Shams University Hospital. J Pure Appl Microbiol. 2020;14(4):2807-2813. doi: 10.22207/JPAM.14.4.56 

 (b)  Paterson GK, Morgan FJ, Harrison EM, Cartwright EJ, Török ME, Zadoks RN, Parkhill J, Peacock SJ, Holmes MA. Prevalence and characterization of human mecC methicillin-resistant Staphylococcus aureus isolates in England. J Antimicrob Chemother. 2014 Apr;69(4):907-10. doi: 10.1093/jac/dkt462. Epub 2013 Nov 27. PMID: 24284779; PMCID: PMC3956372.

References 37 and 38 have been cited in the revised manuscript.

Round 2

Reviewer 1 Report

I have no other recommendations.